# DATA AUGMENTATION FOR DEEP LEARNING BASED ACCELERATED MRI RECONSTRUCTION

## ABSTRACT

Deep neural networks have emerged as very successful tools for image restoration and reconstruction tasks. These networks are often trained end-to-end to directly reconstruct an image from a noisy or corrupted measurement of that image. To achieve state-of-the-art performance, training on large and diverse sets of images is considered critical. However, it is often difficult and/or expensive to collect large amounts of training images. Inspired by the success of Data Augmentation (DA) for classification problems, in this paper, we propose a pipeline for data augmentation for image reconstruction tasks arising in medical imaging and explore its effectiveness at reducing the required training data in a variety of settings. We focus on accelerated magnetic resonance imaging, where the goal is to reconstruct an image from a few under-sampled linear measurements. Our DA pipeline is specifically designed to utilize the invariances present in medical imaging measurements as naive DA strategies that neglect the physics of the problem fail. We demonstrate the effectiveness of our data augmentation pipeline by showing that for some problem regimes, DA can achieve comparable performance to the state of the art on the fastMRI dataset while using significantly fewer training data. Specifically, for 8-fold acceleration we achieve performance comparable to the state of the art with only 10% of the training data for multi-coil reconstruction and with only 33% of the training data for single-coil reconstruction. Our findings show that in the low-data regime DA is beneficial, whereas in the high-data regime it has diminishing returns.

## 1 INTRODUCTION

In magnetic resonance imaging (MRI), an extremely popular medical imaging technique, it is common to reduce the acquisition time by subsampling the measurements, because this reduces cost and increases accessibility of MRI to patients. However, since due to the subsampling, there are fewer equations than unknowns, the signal is not uniquely identifiable from the measurements. To overcome this challenge there has been a flurry of activity over the last decade aimed at utilizing prior knowledge about the signal, in a research area referred to as *compressed sensing* (Candes et al., 2006; Donoho, 2006).

Classical compressed sensing methods reduce the required number of measurements by utilizing prior knowledge about the images during the reconstruction process, often via a convex regularization that enforces sparsity in an appropriate transformation of the image. More recently, deep learning techniques have been used to enforce much more nuanced forms of prior knowledge (see Ongie et al. (2020) and references therein for an overview). The most successful of these approaches aim to directly learn the inverse mapping from the measurements to the image by training on a large set of training data consisting of signal/measurement pairs. This approach often enables faster reconstruction of images, but more importantly, deep learning techniques yield significantly higher quality reconstructions. This in turn enables fewer measurements further reducing image acquisition times. For instance, in an accelerated MRI competition known as FastMRI Challenge (Zbontar et al., 2018), all the top contenders used deep learning reconstruction techniques.

Contrary to classical compressive sensing approaches, however, deep learning techniques typically rely on large sets of training data often consisting of images along with the corresponding measurement. This is also true about the use of deep learning techniques in other areas such as computer vision and Natural Language Processing (NLP) were superb empirical success has been observed.

While large datasets have been harvested and carefully curated by tech companies in areas such as vision and NLP, this is not feasible in many scientific applications including MRI. It is difficult and expensive to collect the necessary datasets for a variety of reasons, including patient confidentiality requirements, cost and time of data acquisition, lack of medical data compatibility standards, and the rarity of certain diseases.

A common strategy to reduce reliance on training data in classification tasks is data augmentation. In a classification setting data augmentation consists of adding additional synthetic data obtained by performing *invariant* alterations to the data (e.g. flips, translations, or rotations) which do not affect the labels. Such data augmentation techniques are commonly used in classification tasks to significantly increase the performance on standard benchmarks such as ImageNet and CIFAR-10. More specific to medical imaging, data augmentation techniques have been successfully applied to registration, classification and segmentation of medical images. More recently, several studies (Zhao et al., 2020b; Karras et al., 2020; Zhao et al., 2020a) have demonstrated that data augmentation can significantly reduce the data needed for GAN training for high quality image generation. In regression tasks such as image reconstruction, however, data augmentation techniques are less common and much more difficult to design in part due to the lack of the aforementioned invariances (e.g., measurements of a rotated image are not the same as measurements from the original image).

The goal of this paper is to explore the benefits of data augmentation techniques for accelerated MRI with limited training data. By carefully taking into account the physics behind the MRI acquisition process we design a data augmentation pipeline, which we call MRAugment, that can successfully reduce the amount of training data required. Specifically, our contributions are as follows:

- We propose a data augmentation technique tailored to the physics of the MR reconstruction problem. We note that it is not obvious how to perform data augmentation in the context of inverse problems, because by changing an image to enlarge the training set, we do not automatically get a corresponding measurement, contrary to classification problems, where the label is retained. In fact, we demonstrate that naive forms of data augmentation that do not properly take into account the underlying physics do not work.

- We demonstrate the effectiveness of MRAugment on a benchmark accelerated MRI data set, specifically on the fastMRI (Zbontar et al., 2018) dataset. For 8-fold acceleration and multi-coil measurements (multi-coil measurements are the standard acquisition mode for clinical practice) we can achieve performance comparable to the state of the art with only $10\%$ of the training data. Similarly, again for 8-fold acceleration and single-coil experiments (an acquisition mode popular for experimentation) MRAugment can achieve the performance of reconstructions methods trained on the entire data set while using only 33% of the training data.

- We perform experiments showing that in a low-data regime, where we have only limited training data MRAugment is beneficial, whereas in the high-data regime it has diminishing returns. In particular we demonstrate that in a low data regime ($\approx 1\%$ of training data), data augmentation very significantly boosts reconstruction performance and captures diagnostically significant detail that is missed without data augmentation. In a moderate data regime ($\approx 10 - 33\%$ of training data), MRAugment still achieves significant improvement in reconstruction performance and may help to avoid hallucinations caused by overfitting without the use of DA.

- Finally, while we focus on the top performing neural networks, MRAugment can seamlessly integrate with any deep learning model and therefore it can be useful in a variety of MRI problems.

## 2 BACKGROUND AND PROBLEM FORMULATION

In this section, we provide a brief background on accelerated MRI and formulate the problem.

MRI is a medical imaging technique that exploits strong magnetic fields to form images of the anatomy. MRI is a prominent imaging modality in diagnostic medicine and biomedical research because it does not expose patients to ionizing radiation, contrary to competing technologies such as computed and positron emission tomography (CT and PET).

However, performing an MR scan is time intensive, which is problematic for the following reasons. First, patients are exposed to long acquisition times in a confined space with high noise levels. Second, long acquisition times induce reconstruction artifacts caused through patient movement,

which sometimes requires patient sedation in particular in pediatric MRI (Vasanawala et al., 2010). Reducing the acquisition time can therefore increase both the accuracy of diagnosis and patient comfort. Furthermore, decreasing the acquisition time needed allows more patients to receive a scan using the same machine. This can significantly reduce patient cost, since each MRI machine comes with a high cost to maintain and operate.

Since the invention of MR in the 1980s there has been tremendous research focusing on reducing their acquisition time. The two main ideas are to i) perform multiple acquisitions simultaneously (Sodickson & Manning, 1997; Pruessmann et al., 1999; Griswold et al., 2002) and to ii) subsample the measurements, known as accelerated acquisition or compressive sensing (Lustig et al., 2008). Most modern scanners combine both techniques, and therefore we consider such a setup.

## 2.1 Accelerated MRI acquisition

In magnetic resonance imaging, measurements of a patient's anatomy are acquired in the Fourier-domain, also called *k-space*, through receiver coils. In the *single-coil* acquisition mode, the k-space measurement $\boldsymbol{k} \in \mathbb{C}^n$ of a complex-valued ground truth image $\boldsymbol{x}^* \in \mathbb{C}^n$ is given by

$$\boldsymbol{k} = \mathcal{F}\boldsymbol{x}^* + \boldsymbol{z},$$

where $\mathcal{F}$ is the two-dimensional Fourier-transform, and $\boldsymbol{z} \in \mathbb{C}^n$ denotes additive noise arising in the measurement process. In parallel MR imaging, multiple receiver coils are used, each of which captures a different region of the image, represented by a complex-valued sensitivity map $\boldsymbol{S}_i$. In this *multi-coil* setup, coils acquire k-space measurements modulated by their corresponding sensitivity maps:

$$\boldsymbol{k}_i = \mathcal{F}\boldsymbol{S}_i\boldsymbol{x}^* + \boldsymbol{z}_i, \quad i = 1, .., N,$$

where $N$ is the number of coils. Obtaining fully-sampled k-space data is time-consuming, and therefore in accelerated MRI we decrease the number of measurements by undersampling in the Fourier-domain. This undersampling can be represented by a binary mask $\boldsymbol{M}$ that sets all frequency components not sampled to zero:

$$\tilde{\boldsymbol{k}}_i = \boldsymbol{M}\boldsymbol{k}_i, \quad i = 1, .., N.$$

We can write the overall forward map concisely as

$$\tilde{\boldsymbol{k}} = \mathcal{A}\left(\boldsymbol{x}^*\right),$$

where $\mathcal{A}\left(\cdot\right)$ is the linear forward operator and $\tilde{\boldsymbol{k}}$ denotes the undersampled coil measurements stacked into a single column vector. The goal in accelerated MRI reconstruction is to recover the image $\boldsymbol{x}^*$ from the set of k-space measurements $\tilde{\boldsymbol{k}}$. Note that—without making assumptions on the image $\boldsymbol{x}^*$—it is in general impossible to perfectly recover the image, because we have fewer measurements than variables to recover. This recovery problem is known as compressive sensing. To make image recovery potentially possible, recovery methods make structural assumptions about $\boldsymbol{x}^*$, such that it is sparse in some basis or implicitly that it looks similar to images from the set of training images.

## 2.2 Traditional accelerated MRI reconstruction methods

Traditional compressed sensing recovery methods for accelerated MRI are based on assuming that the image $\boldsymbol{x}^*$ is sparse in some dictionary, for example the wavelet transform. Recovery is then posed as a typically convex optimization problem:

$$\hat{\boldsymbol{x}} = \arg\min_{\boldsymbol{x}} \left\|\mathcal{A}\left(\boldsymbol{x}\right) - \tilde{\boldsymbol{k}}\right\|^2 + \mathcal{R}(\boldsymbol{x}),$$

where $\mathcal{R}(\cdot)$ is a regularizer enforcing sparsity in a certain domain. Typical functions used in CS based MRI reconstruction are $\ell_1$-wavelet and total-variation regularizers. These optimization problems can be numerically solved via iterative gradient descent based methods.

## 2.3 Deep learning based MRI reconstruction methods

In recent years, several deep learning algorithms have been proposed and convolutional neural networks established new state-of-the-art in MRI reconstruction significantly surpassing the classical

baselines. Encoder-decoder networks such as the U-Net (Ronneberger et al., 2015) and its variants were successfully used in various medical image reconstruction (Hyun et al., 2018; Han & Ye, 2018) and segmentation problems (Çiçek et al., 2016; Zhou et al., 2018). These models consist of two sub-networks: the encoder repeatedly filters and downsamples the input image with learned convolutional filters resulting in a concise feature vector. This low-dimensional representation is then fed to the decoder consisting of subsequent upsampling and learned filtering operations. Another approach that can be considered a generalization of iterative compressed sensing reconstructions consists of unrolling gradient descent iterations and mapping them to a cascade of sub-networks (Zhang & Ghanem, 2018). Several variations of this unrolled method have been proposed recently for MR reconstruction, such as i-RIM (Putzky & Welling, 2019), Adaptive-CS-Net (Pezzotti et al., 2019), Pyramid Convolutional RNN (Wang et al., 2019) and E2E VarNet (Sriram et al., 2020), which is at the top of the fastMRI Challenge Leaderboard at the time of the writing of this paper.

Another line of work, inspired by the deep image prior (Ulyanov et al., 2018) focuses on using the inductive bias of convolutional networks to perform reconstruction without any training data (Jin et al., 2019; Darestani & Heckel, 2020; Heckel & Soltanolkotabi, 2020; Heckel & Hand, 2019; Van Veen et al., 2018). Those methods do perform significantly better than classical un-trained networks, but do not perform as well as neural networks trained on large sets of training data.

## 3 MRAugment: A Data Augmentation Pipeline For MRI

In this section we propose our data augmentation technique, MRAugment, for MRI reconstruction. We emphasize that data augmentation in this setup and for inverse problems in general is substantially different from DA for classification problems. For classification tasks, the label of the augmented image is trivially the same as that of the original image, whereas for inverse problems we have to generate both an augmented target image and the corresponding measurements. This is non-trivial as it is critical to match the noise statistics of the augmented measurements with those in the dataset.

We are given training data in the form of fully-sampled MRI measurements in the Fourier domain, and our goal is to generated new training examples consisting of a subsampled k-space measurement along with a target image. MRAugment is model-agnostic in that the generated augmented training example can be used with any machine learning model and therefore can be seamlessly integrated with existing reconstruction algorithms for accelerated MRI, and potentially beyond MRI.

Our data augmentation pipeline, illustrated in Figure 1, generates a new example consisting of a subsampled k-space measurement $\tilde{k}_a$ along with a target image $\bar{x}_a$ as follows. We are given training examples as fully-sampled k-space slices, which we stack into a single column vector $k = \mathrm{col}(k_1, k_2, ..., k_N)$ for notational convenience. From these, we obtain the individual coil images by applying the inverse Fourier transform as $x = \mathcal{F}^{-1}k$. We generate augmented individual coil images with an augmentation function $\mathcal{D}$, specified later, as $x_a = \mathcal{D}(x)$. From the augmented images, we generate an undersampled measurement by applying the forward model as $\tilde{k}_a = \mathcal{A}(x_a)$. Both $x$ and $x_a$ are *complex-valued*: even though the MR scanner obtains measurements of a real-valued object, due to noise the inverse Fourier-transform of the measurement is complex-valued. Therefore the augmentation function has to generate complex-valued images, which adds an extra layer of difficulty compared to traditional data augmentation techniques pertaining to real-valued images (see Section 3.1 for further details). Finally, the real-valued ground truth image is obtained by combining the coil images $x_{a,i}$ by pixel-wise root-sum-squares (RSS) followed by center-cropping $\mathcal{C}$:

$$\bar{x}_a = \mathcal{C}\left(\mathrm{RSS}(x_a)\right) = \mathcal{C}\left(\sqrt{\sum_{i=1}^{N} |x_{a,i}|^2}\right).$$

In the following two sections we first argue why we generate individual coil images with the data augmentation function, and then discuss the design of the data augmentation function $\mathcal{D}$ itself.

### 3.1 Data augmentation needs to preserve noise statistics

As mentioned before, we are augmenting complex-valued, noisy images. This noise enters in the measurement process when we obtain the fully-sampled measurement of an image $x^*$ as $k = \mathcal{F}x^* + z$, and is well approximated by i.i.d complex Gaussian noise, independent in the real and imaginary

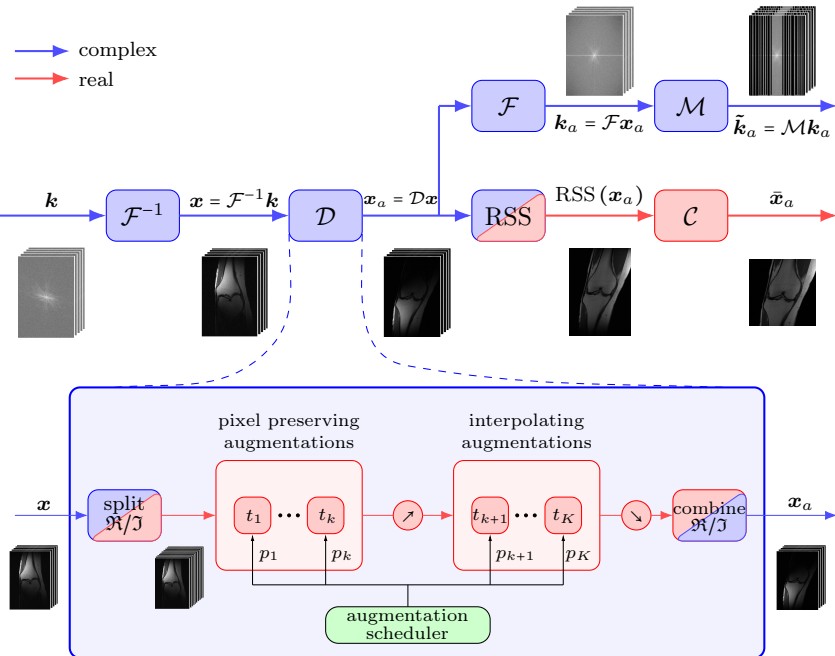

Figure 1: Flowchart of MRAugment, our data augmentation pipeline for MRI.

parts of each pixel (Nishimura, 1996). Therefore, we can write $x = x^* + z'$ where $z'$ has the same distribution as $z$ due to $\mathcal{F}$ being unitary. Since the noise distribution is characteristic to the instrumental setup (in this case the MR scanner and the acquisition protocol), assuming that the training and test images are produced by the same setup, it is important that the augmentation function preserves the noise distribution of training images as much as possible. Indeed, a large mismatch between training and test noise distribution leads to poor generalization (Knoll et al., 2019).

Let us demonstrate why it is non-trivial to generate augmented measurements for MRI through a simple example. A natural but perhaps naive approach for data augmentation is to augment the real-valued target image $\bar{x}$ instead of the complex valued $x$. This would allow us to directly obtain real augmented images from a real target image just as in typical data augmentation. However, this approach leads to different noise distribution in the measurements compared to the test data due to the non-linear mapping from individual coil images to the real-valued target and works poorly as demonstrated in Section 4.

In contrast, if we augment the individual coil images $x$ directly with a linear function $\mathcal{D}$, which is our main focus here, we obtain the augmented k-space data

$$k_a = \mathcal{F}\mathcal{D}x = \mathcal{F}\mathcal{D}(x^* + z') = \mathcal{F}\mathcal{D}x^* + \mathcal{F}\mathcal{D}z',$$

where $\mathcal{F}\mathcal{D}x^*$ represents the augmented signal and the noise $\mathcal{F}\mathcal{D}z'$ is still additive complex Gaussian. A key observation is that in case of transformations such as translation, horizontal and vertical flipping and rotation the noise distribution is exactly preserved. Moreover, for general linear transformations the noise is still Gaussian in the real and imaginary parts of each pixel. This example motivates our choice to i) augment complex-valued images directly derived from the original k-space measurements and ii) consider simple transformations which preserve the noise distribution. Next we overview the types of augmentations we propose in line with these key observations.

### 3.2 TRANSFORMATIONS USED FOR DATA AUGMENTATION

We apply the following two types of image transformation $\mathcal{D}$ in our data augmentation pipeline:
**Pixel preserving augmentations**, that do not require any form of interpolation and simply result in a permutation of pixels over the image. Such transformations are vertical and horizontal flipping, translation by integer number of pixels and rotation by multiples of $90°$. As we pointed out in Section 3.1, these transformations do not affect the noise distribution on the measurements and therefore are

suitable for problems where training and test data are expected to have similar noise characteristics. **General affine augmentations**, that can be represented by an affine transformation matrix and in general require resampling the transformed image at the output pixel locations. Augmentations in this group are: translation by arbitrary (not necessarily integer) coordinates, arbitrary rotations, scaling and shearing. Scaling can be applied along any of the two spatial dimensions. We differentiate between isotropic scaling, in which the same scaling factor $s$ is applied in both directions ($s > 1$: *zoom-in*, $s < 1$: *zoom-out*) and anisotropic scaling in which different scaling factors $(s_x, s_y)$ are applied along different axes.

Figure 2 provides a visual overview of the types of augmentations applied in this paper. Numerous other forms of transformation may be used in this framework such as exposure and contrast adjustment, image filtering (blur, sharpening) or image corruption (cutout, additive noise). However, in addition to the noise considerations mentioned before that have to be taken into account, some of these transformations are difficult to define for complex-valued images and may have subtle effects on image statistics. For instance, brightness adjustment could be applied to the magnitude image, the real part only or both real and imaginary parts, with drastically different effects on the magnitude-phase relationship of the image. That said, we hope to incorporate additional augmentations in our pipeline in the future after a thorough study of how they affect the noise distribution.

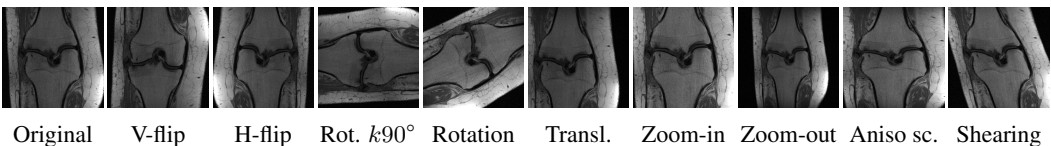

Original   V-flip   H-flip   Rot. $k90°$   Rotation   Transl.   Zoom-in   Zoom-out   Aniso sc.   Shearing

Figure 2: Transformations used in MRAugment applied to a ground truth slice.

### 3.3 SCHEDULING AND APPLICATION OF DATA AUGMENTATIONS

With the different components in place we are now ready to discuss the scheduling and application of the augmentations, as depicted in the bottom half of Figure 1. Recall that MRAugment generates a target image $\bar{x}_a$ and corresponding under-sampled k-space measurement $\tilde{k}_a$ from a full k-space measurement. Which augmentation is applied and how frequently is determined by a parameter $p$, the common parameter determining the probability of applying a transformation to the ground truth image during training, and the weights $\mathcal{W} = (w_1, w_2, ..., w_K)$ pertaining to the $K$ different augmentations, controlling the weights of transformations relative to each other. We apply a given transformation $t_i$ with probability $p_i = p \cdot w_i$. The augmentation function is applied to the coil images, specifically the same transformation is applied with the same parameters to the real and imaginary parts $(\Re\{x_1\}, \Im\{x_1\}, \Re\{x_2\}, \Im\{x_2\}, ..., \Re\{x_N\}, \Im\{x_N\})$ of coil images. If a transformation $t_i$ is sampled (recall that we select them with probabilities $p_i$), we randomly select the parameters of the transformation from a pre-defined range (for example, rotation angle in $[0, 180°]$). To avoid aliasing artifacts, before transformations that require interpolation we first upsample the image and apply the transformations to the upsampled image. Then the result is downsampled to the original size.

A critical question is how to schedule $p$ over training in order to obtain the best model. Intuitively, in initial stages of training no augmentation is needed, since the model can learn from the available (possibly limited) original training data. As training progresses the network learns to fit the original data points and their utility decreases over time. We found schedules starting from $p = 0$ and increasing over epochs to work best. The ideal rate of increase depends on both the model size and amount of available training data. We experimented with linear (ramp) and exponential scheduling and observed that linear functions work best when we have a large amount of training data compared to the model size. In contrast, steeper exponential scheduling helps in the low-data regime.

## 4 EXPERIMENTS

In this section we explore the effectiveness of MRAugment in the context of accelerated MRI reconstruction in various regimes of available training data sizes on the fastMRI dataset. We present a detailed description of this dataset in Appendix A.

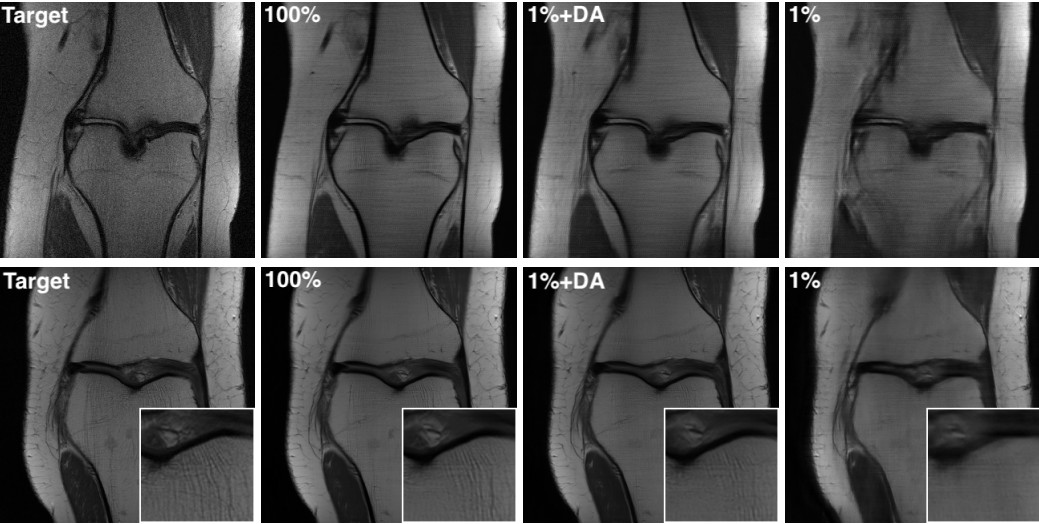

Figure 3: Visual comparison of single-coil (top row) and multi-coil (bottom-row) reconstructions using varying amounts of training data with and without data augmentation. We achieve reconstruction quality comparable to the state of the art but using 1% of the training data. Without DA fine details are completely lost.

**Experimental setup.** We use the state-of-the-art End-to-End VarNet model (Sriram et al., 2020), because it is as of now the best performing neural network on the fastMRI dataset. We measure performance in terms of the structural similarity index measure (SSIM), which is the standard evaluation metric on the fastMRI dataset. We study the performance of MRAugment as a function of the size of the training set. We construct different subsampled training sets by randomly sampling volumes of the original training dataset and adding all slices of the sampled volumes to the new subsampled dataset. The original training set consists of approximately $35k$ MRI slices in $973$ volumes and we subsample to $1\%$, $10\%$, $33\%$ and $100\%$ of the original size. We measure performance on the original validation set separate from the training set. For all experiments, we apply random masks by undersampling whole kspace lines in the phase encoding direction and including $4\%$ of lowest frequency adjacent kspace lines in order to be consistent with baselines in (Zbontar et al., 2018). For both the baseline experiments and for MRAugment, we generate a new random mask for each slice on-the-fly while training by uniformly sampling kspace lines, but use the same fixed mask for each slice within the same volume on the validation set (different across volumes). This technique is standard for models trained on this dataset and not part of our data augmentation pipeline. For augmentation probability scheduling we use

$$p(t) = \frac{p_{max}}{1 - e^{-c}}(1 - e^{-tc/T}),$$ 
(4.1)

where $t$ is the current epoch, $T$ denotes the total number of epochs, $c = 5$ and $p_{max} = 0.6$ for $1\%$ training data experiments and $p_{max} = 0.55$ in other data augmentation experiments. More experimental details can be found in Appendix B and the source code is attached in the supplementary.

**Single-coil experiments.** For single-coil acquisition we are able to exactly match the performance of the model trained on the full dataset using *only a third of the training data* as depicted on the left in Fig. 4. Moreover, with only $10\%$ of the training data we achieve $99.5\%$ of the SSIM trained on the full dataset. The visual difference between reconstructions with and without data augmentation becomes striking in the low-data regime. As seen in the top row of Fig. 3, the model without DA was unable to reconstruct any of the fine details and the results appear blurry with strong artifacts. Applying MRAugment significantly improves reconstruction quality both in a quantitative and qualitative sense, visually approaching that obtained from training on the full dataset but using *hundred times less data*.

**Multi-coil experiments.** As depicted on the right in Fig. 4 for multi-coil acquisition we closely match the state of the art while significantly reducing training data. More specifically, we approach

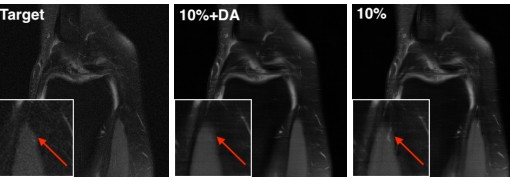

Figure 4: Single-coil (left) and multi-coil (right) validation SSIM vs. # of training images.

Figure 5: Hallucinated features appear on reconstructions without data augmentation.

the state of the art SSIM within $0.6\%$ using $10\%$ of the training data and within $0.25\%$ with $33\%$ of training data. As seen in the bottom row of Fig. 3, when using only $1\%$ of the training data we successfully reconstruct fine details comparable to that obtained from training on the full dataset, while high frequencies are completely lost without DA. More visual comparisons for both single-coil and multi-coil reconstruction can be found in Appendix D.

**Hallucinations.** An unfortunate side-effect of deep learning based reconstructions may be the presence of hallucinated details. This is especially problematic in providing accurate medical diagnosis and lessens the trust of medical practicioners in deep learning. We observed that data augmentation has the potential benefit of eliminating hallucinations by preventing overfitting to training data, as seen in Fig. 5.

Our empirical observations on data augmentation can be summarized as follows:

- In the low data regime ($\approx 1\%$ of training data), data augmentation very significantly boosts reconstruction performance. The improvement is large both in terms of raw SSIM and visual reconstruction quality. Using MRAugment, fine details are recovered that are completely missing from reconstructions without DA. This suggests that DA improves the value of reconstructions for medical diagnosis, since health experts typically look for small features of the anatomy.

- In the moderate data regime ($\approx 10 - 33\%$ of training data), MRAugment still achieves significant improvement in reconstruction SSIM. We want to emphasize the significance of seemingly small differences in SSIM close to the state of the art and invite the reader to visit the FastMRI Challenge Leaderboard that demonstrates how close the best performing models are. Moreover, MRAugment achieves visual improvement in fine details that may have diagnostic value. Fig. 5 shows that in some cases hallucinated details can be avoided in this regime using data augmentation. Additional visual comparison of recovered details is provided in Appendix D.

- Data augmentation applied to the full training dataset has diminishing returns. It does not improve performance if applied to the full dataset, but it does not degrade it either. This suggests that it may be possible to further improve upon the state of the art when using the entire dataset by combining data augmentation with more advanced transformations and larger models. We plan to pursue this exciting direction in our future work.

**Ablation studies.** We performed ablation studies on $1\%$ training data in order to better understand which augmentations are useful. We use the multi-coil experiment with all augmentations as baseline and tune augmentation probability for other experiments such that the probability that a certain slice is augmented by at least one augmentation is the same across all experiments. We depict results on the validation dataset in Table 1. Both pixel preserving and general (interpolating) affine transformations are useful and can significantly increase reconstruction quality. Furthermore, we observe that their effect is complementary: they are helpful separately, but we achieve peak reconstruction SSIM when all applied together. Finally, the utility of pixel preserving augmentations seems to be lower than that of general affine augmentations, however they come with a negligible additional computational cost.

Furthermore, we investigate the effect of varying the augmentation probability scheduling function. The results on the validation dataset are depicted in Table 2, where *exponential, $\hat{p}$* denotes the exponential scheduling function in (4.1), with $p_{max} = \hat{p}$ and *constant, $\hat{p}$* means we use a fixed augmentation probability $\hat{p}$ throughout training. We observe that scheduling starting from low augmentation probability and gradually increasing is better than a constant probability, as initially the network does not benefit much from data augmentation as it can still learn from the original samples. Furthermore, too low or too high augmentation probability both degrade performance. If the

| Augmentations | Val. SSIM |
|---|---|
| none | 0.8396 |
| pixel preserving only | 0.8585 |
| interpolating only | 0.8731 |
| all augmentations | **0.8758** |

| Aug. scheduling | Val. SSIM |
|---|---|
| none | 0.8396 |
| exponential, 0.3 | 0.8565 |
| constant, 0.3 | 0.8588 |
| exponential, 0.6 | **0.8758** |
| constant, 0.6 | 0.8611 |
| exponential, 0.8 | 0.8600 |

Table 1: Comparison of peak validation SSIM applying various sets of augmentations on 1% of training data, multi-coil acquisition.

Table 2: Comparison of peak validation SSIM using different augmentation probability schedules on 1% of training data, multi-coil acquisition.

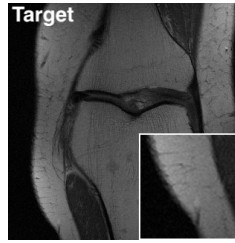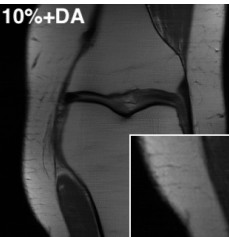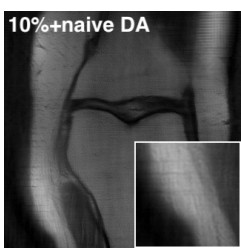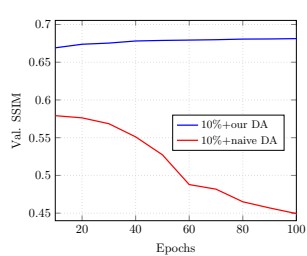

(a) Naive data augmentation that does not preserve measurement noise distribution leads to significantly degraded reconstruction quality.

(b) Naive data augmentation degrades generalization performance.

Figure 6: Experimental results comparing MRAugment with naive data augmentation.

augmentation probability is too low, the network may overfit to training data as more regularization is needed. On the other hand, too much data augmentation hinders reconstruction performance as the network rarely sees images close to the original training distribution.

**Noise considerations.** Finally, we would like to emphasize the importance of applying DA in a way that takes into account the measurement noise distribution. When applied incorrectly, DA leads to significantly worse performance than not using any augmentation. We train a model using DA without considering the measurement noise distribution as described in Section 3.1 by augmenting real-valued target images. We use the same exponential augmentation probability scheduling for MRAugment and the naive approach. As Fig. 6b demonstrates, reconstruction quality degrades over training using the naive approach. This is due to the fact that as augmentation probability increases, the network is being fed less and less original images, whereas the poorly augmented images are detrimental for generalization due to the mismatch in train and validation noise distribution. On the other hand, MRAugment clearly helps and validation performance steadily improves over epochs. Fig. 6a provides a visual comparison of reconstructions using naive DA and our data augmentation method tailored to the problem. Naive DA reconstruction exhibits high-frequency artifacts and low image quality caused by the mismatch in noise distribution. These drastic differences underline the vital importance of taking a careful, physics-informed approach to data augmentation for MR reconstruction.

## 5 CONCLUSION

In this paper, we develop a physics-based data augmentation pipeline for accelerated MR imaging. We find that DA yields significant gains in a low-data regime which can be beneficial in applications where only little training data is available or where the training data changes quickly. We also demonstrate that even with a moderate amount of training data, DA enables reconstructions that contain diagnostically significant detail and may help avoiding hallucinations caused by overfitting without DA.

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

## A  THE FASTMRI DATASET

The fastMRI dataset (Zbontar et al., 2018) is a large open dataset of knee and brain MRI volumes. The train and validation splits contain fully-sampled k-space volumes and corresponding target reconstructions for both (simulated) single-coil and multi-coil acquisition. The knee MRI dataset we are focusing on in this paper includes 973 train volumes (34742 slices) and 199 validation volumes (7135 slices). The target reconstructions are fixed size $320{\times}320$ center cropped images corresponding to the fully-sampled data of varying sizes. The under-sampling ratio is either $25\%$( $4\times$ acceleration) or $12.5\%$ ($8\times$ acceleration). Under-sampling is performed along the phase encoding dimension in k-space, that is columns in k-space are sampled. A certain neighborhood of adjacent low-frequency lines are always included in the measurement. The size of this fully-sampled region is $8\%$ of all frequencies in case of $4\times$ acceleration and $4\%$ in case of $8\times$ acceleration.

## B  EXPERIMENTAL DETAILS

Here we provide details of the setup for our main experiments.

**Datasets.** We use the FastMRI (Zbontar et al., 2018) single-coil and multi-coil knee dataset for our experiments. For creating the sub-sampled datasets, we uniformly sample volumes from the training set, and add all slices from the sampled volumes. Our validation results are reported on the whole validation dataset. Images in the dataset have varying dimensions. Due to GPU memory considerations we center-cropped the input images to $640 \times 368$ pixels (which covers most of the images). We use random undersampling masks with $8\times$ acceleration and $4\%$ fully-sampled low-frequency band, undersampled in the phase encoding direction by masking whole kspace lines. We generate a new random mask for each slice on-the-fly while training, but use the same fixed mask for each slice within the same volume on the validation set (different across volumes).

**Model.** We train the default E2E-VarNet network from Sriram et al. (2020) with 12 cascades (approx. $30M$ parameters) for both the single-coil and multi-coil reconstruction problems. For single-coil data we remove the Sensitivity Map Estimation sub-network as sensitivity maps are not relevant in this problem.

**Hyperparameters and training.** We use an Adam optimizer with 0.0003 learning rate following Sriram et al. (2020). We train the baseline model on the full training dataset for 50 epochs. For the smaller, sub-sampled datasets we train for the same computational cost as the baseline, that is we train for $N \cdot 50$ epochs on $1/N$th of the training data. Without data augmentation, we observe a saturation in validation SSIM during this time. With data augmentation we trained $50\%$ longer as we still observe improvement in validation performance after the standard number of epochs. We report the best SSIM on the validation set throughout training. We train on 4 GPUs for single-coil data and on 8 GPUs for multi-coil data. The batch size matches the number of GPUs used for training, since a GPU can only hold a single datapoint.

**Data augmentation parameters.** The transformations and their corresponding probability weights and ranges of values are depicted in Table 3. We adjust the weights so that groups of transformations such as rotation (arbitrary, by $k \cdot 90°$), flipping (horizontal or vertical) or scaling (isotropic or anisotropic) have similar probabilities. For both the affine transformations and upsampling we use bicubic interpolation. Due to computational considerations we only use upsampling before transformations for the single-coil experiments.

## C  COMPARISON OF ADDITIONAL METRICS

In order to provide more in-depth comparison for our main experiment, here we provide results on PSNR as an additional image quality metric. We observe significant and consistent improvement in PSNR when applying MRAugment (Fig. 7) with similar trends to SSIM: the improvement is the most prominent in the low-data regime, but still significant in the moderate domain.

| Transformations | Range of values | $w_i$ |
|---|---|---|
| Horizontal flip | flipped/not flipped | 0.5 |
| Vertical flip | flipped/not flipped | 0.5 |
| Rotation by $k \cdot 90°$ | $k \in \{0, 1, 2, 3\}$ | 0.5 |
| Rotation | $[-180°, 180°]$ | 0.5 |
| Translation | width: $[-8\%, 8\%]$, height: $[-12.5\%, 12.5\%]$ | 1.0 |
| Isotropic scaling | $[0.75, 1.25]$ | 0.5 |
| Anisotropic scaling | $[0.75, 1.25]$ along each axes | 0.5 |
| Shearing | $[-12.5°, 12.5°]$ | 1.0 |

Table 3: Data augmentation configuration for all experiments.

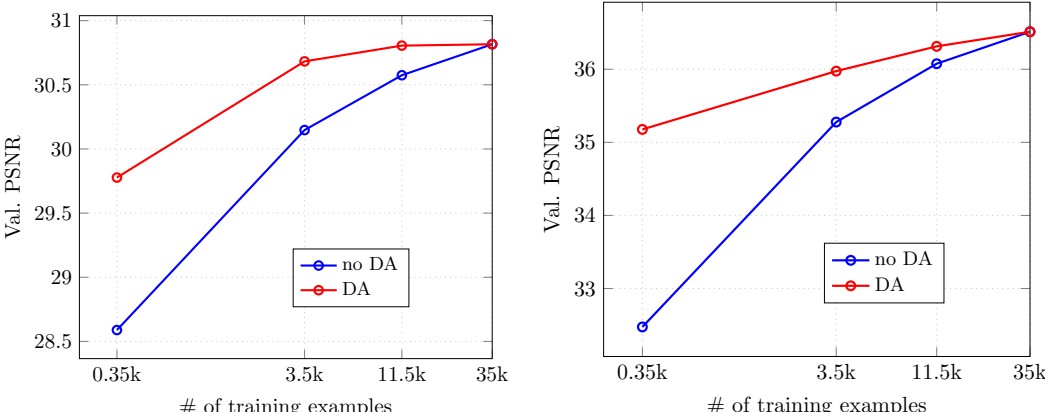

Figure 7: Single-coil (left) and multi-coil (right) validation PSNR vs. # of training images.

## D    MORE EXPERIMENTAL RESULTS

**Additional reconstructions.** In order to demonstrate that MRAugment works well across a wide range of MR slices and the results depicted in Section 4 are not cherry-picked, here we provide additional reconstructions with and without data augmentation. In multi-coil reconstructions the visual differences are more subtle, therefore we magnified regions with fine details for better comparison.

Complete version of Figure 3 extended with reconstructions with and without data augmentation for 10% and 33% training data are presented in Figures 9 and 10 respectively.

Figures 11 and 12 provide more reconstructed slices randomly sampled from the validation dataset with and without DA.

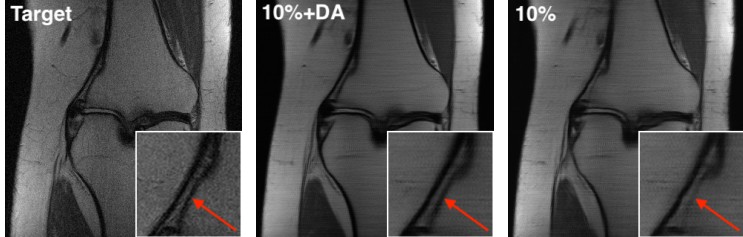

Figure 8: MRAugment recovers additional fine details even in the moderate data regime.

**Other models.** Even though we demonstrated our DA pipeline on E2E-VarNet, the potential of our technique is not limited to a specific model. We performed preliminary experiments on i-RIM Putzky & Welling (2019), another top performing model on single-coil MR reconstruction. We kept the hyperparameters proposed in Putzky & Welling (2019) for the single-coil problem with modifications as follows. Due to computational considerations, we decreased the number of invertible layers to

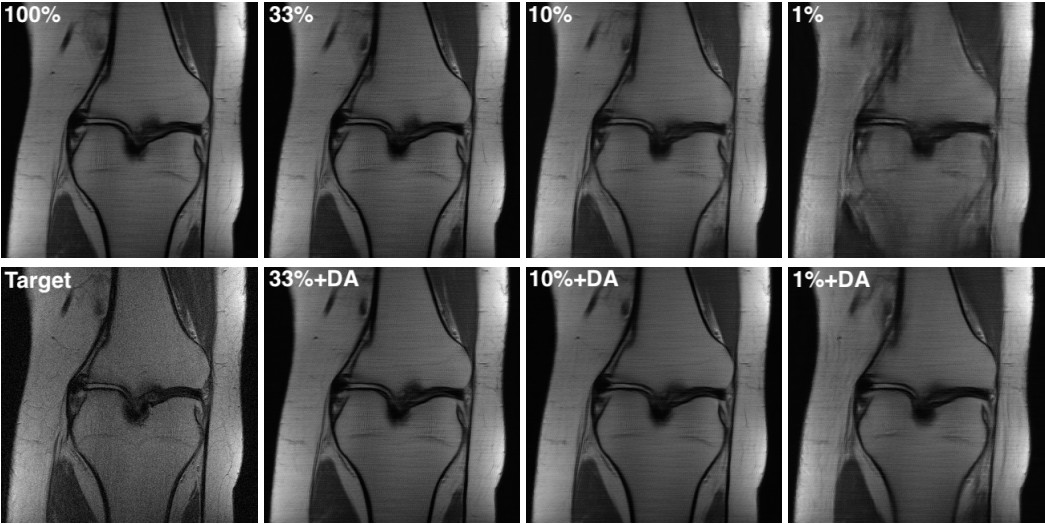

Figure 9: Visual comparison of single-coil reconstructions depicted in Figure 3 extended with additional images corresponding to various amount of training data.

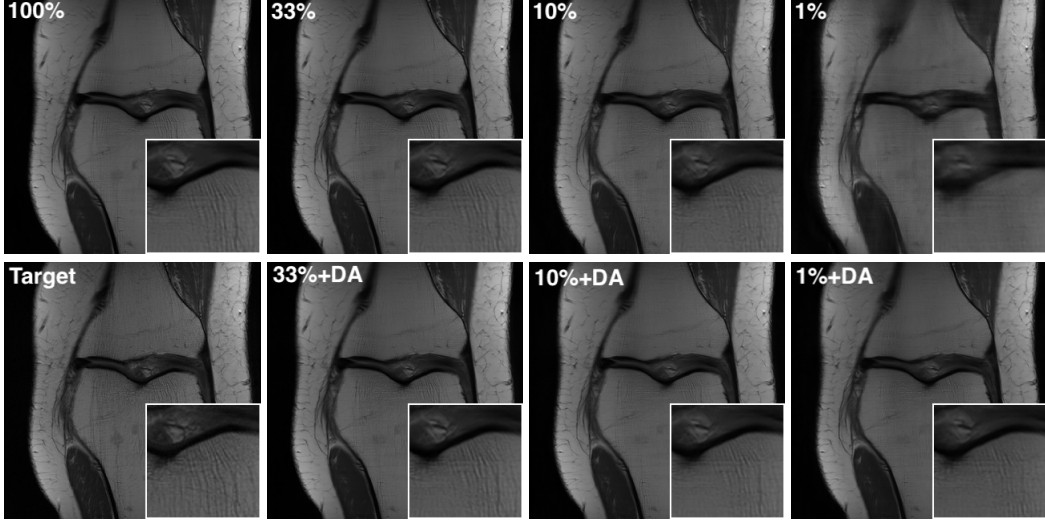

Figure 10: Visual comparison of multi-coil reconstructions depicted in Figure 3 extended with additional images corresponding to various amount of training data.

6 with $[64, 128, 256, 256, 128, 64]$ hidden features inside the reversible blocks and $[1, 2, 4, 4, 2, 1]$ kernel strides in each layer, resulting in a model with $20M$ parameters. In order to further reduce training time, we trained on volumes without fat suppression that take up $50\%$ of the full fastMRI knee dataset. We refer to this new reduced dataset ass 'full' in this section. Finally, we trained on $368x368$ center crops of input images for each experiment. We used ramp scheduling with augmentation probability linearly increasing from 0 to $p_{max} = 0.4$. The acceleration factor and under-sampling mask were the same as in Section 4 . As depicted in Fig. 13 , our experiments show that applying data augmentation to only $10\%$ of the training data can match the performance of the model trained on the full dataset.

| Ground truth | 100% training data | 1% training data + DA | 1% training data |
|---|---|---|---|

Figure 11: Visual comparison of single-coil reconstructions using varying amounts of training data with and without data augmentation.

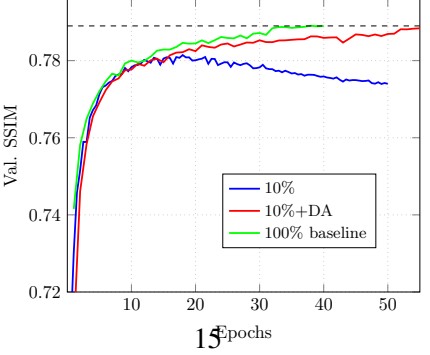

| Ground truth | 100% training data | 1% training data + DA | 1% training data |
|---|---|---|---|

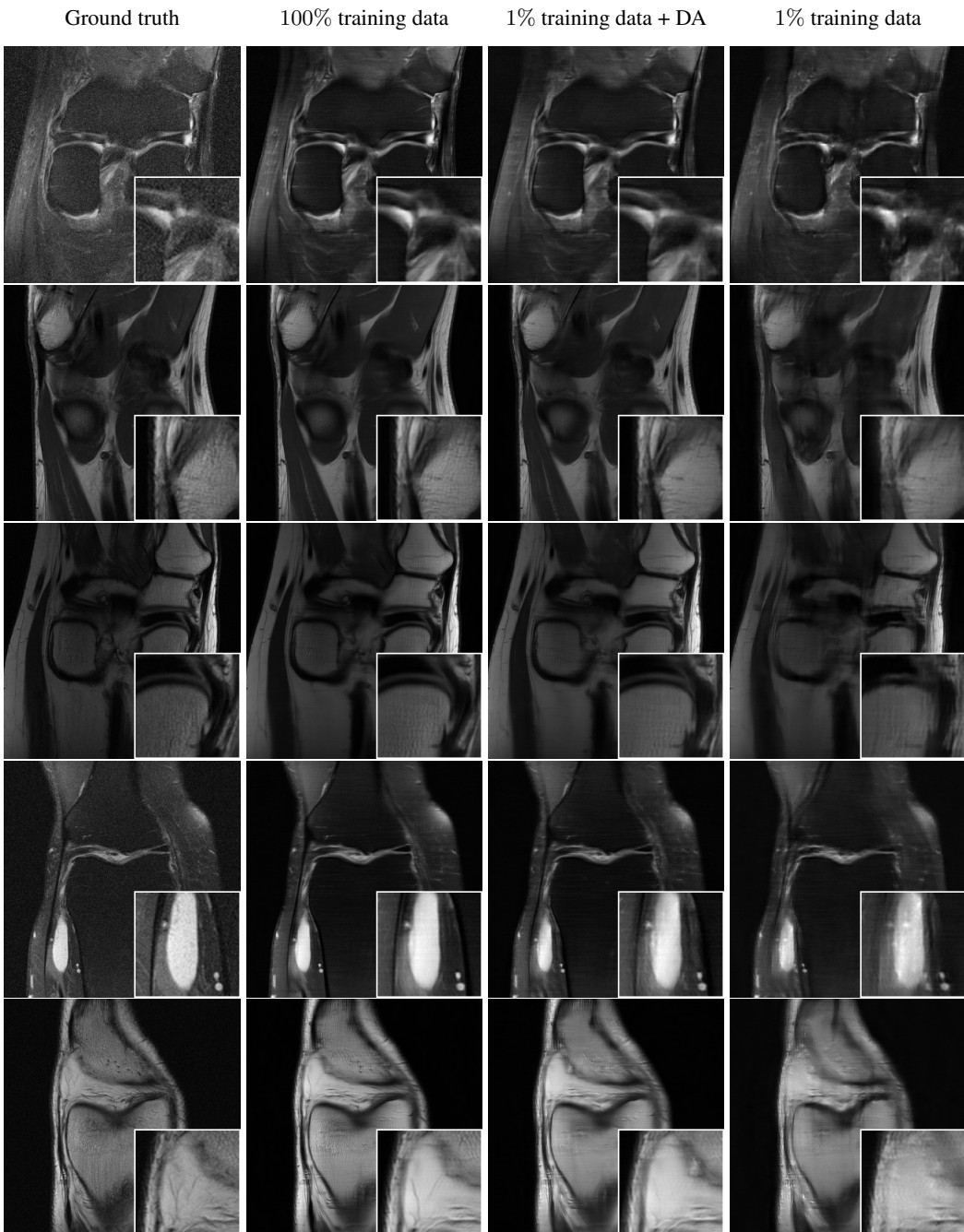

Figure 12: Visual comparison of multi-coil reconstructions using varying amounts of training data with and without data augmentation.

Figure 13: Experimental results on the i-RIM network. We are able to achieve SSIM comparable to the 100% baseline with only 10% of the training data.

# E    SENSITIVITY MAP PRESERVING AUGMENTATIONS

In the multi-coil case, our augmentation pipeline applies transformations to the underlying object modulated by the different coil sensitivity maps. In particular, the fully sampled measurement of the

$i$th coil in the image domain takes the form

$$x_i = S_i x^* + z_i',$$ (E.1)

where $z_i' = \mathcal{F}^{-1} z_i$ is i.i.d Gaussian noise obtained via a unitary transform of the original measurement noise. Assuming linear augmentations, the augmented coil image from MRAugment can be written as

$$x_{a,i} = \mathcal{D}(S_i x^* + z_i') = \mathcal{D} S_i x^* + \mathcal{D} z_i',$$ (E.2)

where the additive noise is still Gaussian and the augmentation is applied to the coil sensitivity maps as well. However, the models we experimented with had no issues learning the proper mapping from transformed sensitivity maps as our experimental results show.

It is natural to ask if data augmentation would be possible by directly augmenting the object. If the coil sensitivities are known or are estimated a priori, one may recover the object from the various coils as

$$x = \sum_{j=1}^{N} S_j^* x_j = \sum_{j=1}^{N} S_j^* (S_j x^* + z_j') = (\sum_{j=1}^{N} S_j^* S_j) x^* + \sum_{j=1}^{N} S_j^* z_j' = x^* + \sum_{j=1}^{N} S_j^* z_j',$$

where $S_j^*$ is the complex conjugate of $S_j$ and $\sum_{j=1}^{N} S_j^* S_j = I$ due to typical normalization (Sriram et al., 2020). Then, we can apply the augmentation as

$$x_a = \mathcal{D} x = \mathcal{D}(x^* + \sum_{i=1}^{N} S_j^* z_j') = \mathcal{D} x^* + \mathcal{D} \sum_{j=1}^{N} S_j^* z_j'.$$

Finally, we obtain the augmented coil images as

$$x_{a,i} = S_i x_a = S_i \mathcal{D} x^* + S_i \mathcal{D} \sum_{j=1}^{N} S_j^* z_j'.$$ (E.3)

Comparing this result to E.1, one may see that the additive noise has a very different distribution from the original noise distribution, even worse noise on different augmented coil images are now correlated! Moreover, in this last step we modulated the additive noise with the sensitivity map, resulting in regions with low noise variance where the coil is less sensitive and higher variance in more sensitive regions. This does not reflect the i.i.d noise distribution present on the original coil measurements. Finally, the sensitivity maps are typically not known and need to be estimated before we can apply this augmentation technique, which can introduce additional inaccuracies in the augmentation pipeline.

## F ROBUSTNESS

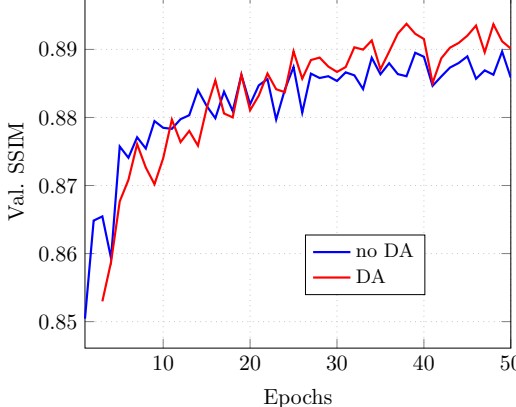

Figure 14: Generalization performance of models trained on knee MRI data and evaluated on brain MRI data. Data augmentation improves validation performance on unseen datasets.

We observed an additional benefit of data augmentation, namely its potential to improve the robustness of the trained model. First, we trained a VarNet model on the complete fastMRI knee train dataset using the hyperparameters recommended in Sriram et al. (2020), and evaluated the network on the fastMRI brain validation dataset throughout training. We repeated the experiment with the same hyperparameters, but with MRAugment turned on (exponential scheduling, peak probability $0.4$). The results can be seen in Fig. 14. The regularizing effect of data augmentation impedes the network to overfit to the training dataset, thus the resulting model is more robust to shifts in test distribution. This results in higher reconstruction quality in terms of SSIM on unseen brain data when using MRAugment.

