# OpenReview forum: "Data augmentation for deep learning based accelerated MRI reconstruction"
_ICLR.cc/2021/Conference — Reject_

### Official Review · AnonReviewer1 · 2020-10-27
**The authors propose a data augmentation pipeline specifically for MRI reconstruction.**

**Rating:** 4
**Confidence:** 5

**Review:**


In this paper, the authors design a data-augmentation pipeline for the domain of MRI reconstruction (specifically, by proposing sensible guidelines for augmenting k-space data when learning image reconstructions, to preserve the noise characteristics of the image data). They show that this pipeline works as you might expect data augmentation to work: it boosts results for small training sets and becomes increasingly less effective as the training set grows. However, while the problem domain is of interest, there are issues with the presented work.

One of the most prominent weaknesses of the work as a whole is that there is little attempt to compare to sensible baselines. Table 1, with the chosen augmentation parameters, does not have any ablation or intuition about why the choices are made. It would be interesting to compare the effects of reducing this fairly large selection of augmentations. How far can we get with just flips and rotations, for instance? While the authors do compare against a 'naive' augmentation, these results are misleading. It isn't 'naive' to apply augmentation techniques from other domains: it is simply wrong!

When the authors evaluate performance with different percentages of the training dataset withheld, the authors must clarify their method for subsampling. Is it random over all slices, or stratified by subject? If it is completely random, the same subject can have slices in both the included and excluded parts of the dataset, even though different slices from the same subject will be highly correlated.

The analysis of results tends to be highly qualitative. The authors point out that 'fine detail' can be obtained with just 1% of the training data, (Figure 3). Similarly, the lack of hallucinated features is not quantified but instead shown with a Figure with one example of reduced hallucination. While the images supporting these two results are visually appealing, there is no quantification of these differences, or attempts to find places where the reconstruction fails. In fact, it is likely that with some searching, instances could be found of hallucinated features, so suggesting that hallucination has been 'eliminated' via augmentation is an overclaim.  Overall, quantifying results of this sort would strengthen the work considerably.

More minor points:
* The analysis is restricted to one dataset (fastMRI), on which it doesn't truly excel — it doesn't boost performance at all when using the full training set. Attempts to source other datasets would be welcome.
* The nomenclature of the paper needs attention: specifically,  compressed/compressive sensing are both used. Stick to one, or mention them both in the introduction as being interchangeable.
* The paper states that MRI does not use radiation on patients: correct this to 'avoids using ionizing radiation'.

---

> ### Author Response · Authors · 2020-11-19
> **Response to Reviewer1 - pt. 1**
>
> We would like to thank the reviewer for the valuable insight and for taking the time reviewing our paper. We would like to address your concerns in detail.
>
> **Re sensible baselines:** We would like to point out that, to the best of our knowledge, data augmentation has not yet been successfully applied to the MRI reconstruction problem in the literature. Therefore, it is not straightforward to compare MRAugment to any sensible baseline. We do not consider naive data augmentation a baseline against our method. We simply use it as a toy example to demonstrate that noise considerations are very important in data augmentation for reconstruction problems, and if we were to apply augmentation as it is widely used in classification problems, the method would fail. Furthermore, we added ablation studies in the Experiments section demonstrating that both pixel preserving and general interpolating affine transformations are useful separately, they are complementary and the best results are achieved by combining them. That said, if you have a specific baseline in mind we are happy to carry out comparisons in future iterations of this paper.
>
> **Re qualitative analysis of results:** We agree that some of our evaluation of reconstruction quality is qualitative, however the basis of our whole analysis is comparison of SSIM metric of reconstructions with and without data augmentation with solid state-of-the-art baselines on the dataset. We think it is important that, beyond showing that data augmentation significantly boosts SSIM, the results also *look* better, thus the qualitative statements. However, we are actively working with a radiologist to get their opinion on the diagnostic value of these improvements. With respect to hallucinations, we provided an intuitive discussion why data augmentation might help eliminate hallucinations via preventing overfitting. However, this is mostly an interesting side-effect of our method and not a main focus of our paper. We changed the language of the paper to reflect your comment on overclaiming the impact on hallucinations. We agree with the reviewer that it would be an interesting future direction to investigate closer what percentage of the hallucinations can be eliminated using data augmentation.
>
> **Re MRAugment not helping on the full dataset:** We would like to point out that the main goal of our paper is to demonstrate how data augmentation can help in the low data regime for MRI reconstruction (which overwhelmingly arises in practice) and *not* to exceed the fastMRI baseline on the full dataset. In fact the reason we used a large dataset such as the fastMRI dataset is to analyze the impact of the amount of training data with and without data augmentation. We want to emphasize that in most MRI applications such a large dataset is not available for training. Therefore, there is a stronger motivation to improve low-data reconstruction performance than improving a baseline for which the amount of training data is rarely  met in practice. A good example of this is low-intensity magnetic field MRIs which show a lot of promise in making MRI safer and applicable to many new settings including patients with pacemakers or metallic inserts but of which there are only three in the entire US dedicated to wide variety of research areas so that it is not feasible to gather such large data sets. Even in the high-intensity regime such as the fastMRI data it is not clear that the conclusions generalize to other MR machines without calibration and it may not be feasible to gather such a large training data for each new machine immediately. That said, we believe that with increased network capacity (add more layers or make the network a bit wider), by using data augmentation we could even improve upon the state-of-the-art as DA is most helpful in a regime where the network capacity is so large that without DA the network overfits. However, this is currently out of the scope of this paper as each experiment on the entire data set takes almost a week to train on our 8-GPU server and therefore overparameterization by a factor of even 4 requires training for almost a month which is currently not feasible for us. However, we are in the process of purchasing more GPUs and we are optimistic that we can eventually improve over the entire data set. Additionally, we do observe a benefit of MRAugment on the full training dataset in the form of improved performance on new datasets not seen by the model. We added further details on this in Appendix F.

---

> > ### Author Response · Authors · 2020-11-19
> > **Response to Reviewer1 - pt. 2**
> >
> > **Re training set subsampling:** The reviewer raised a very good point about sampling for the training datasets. We clarified this in the main part of our paper now, thank you for the comment. To answer your question, we randomly sampled whole volumes from the training dataset, and added all slices from the sampled volumes to our training set. The validation set is completely separate from the training set and is the same for all experiments, therefore it is not possible that some slices from the same volume are present in both training and validation sets.
> >
> > **Regarding other datasets:** the fastMRI dataset is the largest publicly available MRI dataset, allowing us to investigate the effect of the amount of training data on reconstructions and the effect of data augmentation. Moreover, there are highly competitive baseline models for this dataset to provide comparison. On smaller datasets these baselines are much more questionable, and varying the amount of training data in different regimes would be impossible.
> >
> > **Other minor comments:** thank you for pointing out the inconsistency and the wording issue, we fixed those in the new version of the paper!

---

### Official Review · AnonReviewer2 · 2020-10-28
**Physics aware data augmentation for MRI reconstruction**

**Rating:** 5
**Confidence:** 5

**Review:**

This paper presents a method to use data augmentation to improve accelerated MRI reconstruction when the amount of training data is limited. This is an important problem since MRI data is expensive to obtain. Traditional image augmentation methods can't be applied directly for this problem because MR images are complex valued. Further, the applied transformations need to preserve the noise distribution, without which model performance degrades significantly.

The key contribution of the paper is a carefully designed data augmentation policy inspired by the physics of the MR domain. Using this method the author obtains significant improvement in the low data regime for both single-coil and multi-coil reconstruction problems. The proposed method is also independent of the reconstruction model. The results shown in the paper are highly encouraging. The paper is clearly written and is easy to follow.

Some things to clarify / improve the paper:

1. How was the data subsampled? For the 1% experiments, did you choose 1% of slices or 1% of volumes? Since different slices of the same volume have a large amount of redundant information, these two methods are not equivalent. Sampling 1% of volumes is a more realistic scenario.
2. What model was used for single-coil reconstruction? The authors mention using the E2E-VarNet model, but that model was designed specifically for multi-coil data.
3. The paper showed that data augmentation improves SSIM as well as image quality when the amount of training data is small. However, in practice it is important for the model to generalize to different pathologies. It is unclear if the model trained with DA can reconstruct pathologies that were not present in training data. This is important because when the amount of training data is small, most pathologies will be missing from it. Please include examples of pathologies that were not observed during training.
4. The experimental results in the paper clearly demonstrate the benefits of the proposed data augmentation method. However, the paper lacks sufficient ablation experiments that leaves a number of questions unanswered, for example: (a) are all of the augmentations used in the paper required? (b) how does the augmentation schedule proposed by the paper compare to other schedules like uniform augmentation?
5. Did you use random masks during training? Training with random masks (either fully random masks or fixed-width masks with random starting offsets) has the effect of data augmentation without the use of explicit augmentations. If the study used fixed masks, then the effect of DA is probably being exaggerated. It would be good to include additional experiments with random masks.

---

> ### Author Response · Authors · 2020-11-19
> **Response to Reviewer2**
>
> Thank you very much for reviewing our paper, we really appreciate the effort and the useful feedback. We would like to address your questions with respect to our paper.
>
> 1. This is a great point. We randomly sampled whole volumes of the full training dataset and added all slices corresponding to the sampled volumes. As you pointed out, sampling full volumes instead of just slices is very important in order to obtain a balanced dataset and to mimic a real-world scenario. We added details on dataset sampling to the main part of our paper.
> 2. The reviewer is correct that VarNet introduces a sub-network for sensitivity map estimation for multi-coil reconstruction. We simply removed this part of the network on single-coil data. We verified that the single-coil reconstructions obtained from this slightly modified network are comparable to the state-of-the-art. Other parts of the VarNet architecture are not specific to multi-coil data.
> 3. We agree with the reviewer that it is important to see whether MRAugment, beyond visually improving reconstructions,  can help diagnose pathologies in the low data regime that is not possible without data augmentation. However, we would like to point out that pathologies are not annotated in the fastMRI dataset, and therefore this comparison is challenging without further expert guidance. That said, we believe this to be an exciting future direction and are actively working towards gathering such data in a related MR domain.
> 4. We agree that it would be exciting to see extensive experiments on the contribution of different augmentation functions. However, training on the fastMRI knee dataset takes about 1200 GPU hours for a single experiment (without data augmentation) with multi-coil data and since we have access to only academic level of computational resources it takes close to a week for a single experiment. We verified the utility of the different augmentations on a subsampled dataset and observed that they are individually useful and their effect is complementary. We included these ablation studies in the Experiments section. As the reviewer pointed out and we also mentioned in the paper, finding an empirical method to tune the augmentation strength would be very helpful and it is something we are considering as future research. To address the reviewer’s question we added extra experiments in the Experiments section  investigating how augmentation scheduling, including constant functions,  impacts reconstruction quality.
> 5. Thank you for your comment on the undersampling mask, we added details on how the masks were generated in the main part of our paper. In particular, we used randomized masks for both data augmentation and the baseline experiments where we sample full kspace lines in the phase encoding direction, which is easy to implement on a real scanner and is the standard for fastMRI knee baselines.

---

### Official Review · AnonReviewer4 · 2020-10-29
**The authors present a data augmentation framework for MRI reconstruction and evaluate performance on a large scale publicly available data.**

**Rating:** 6
**Confidence:** 4

**Review:**

An augmentation method is proposed for MR image reconstruction, and is shown that significant results can be achieved with a small fraction of training data.

Comments
One of the major concern in image reconstruction problems is the availability of fully sampled data, while the dataset used in the problem provides this, how the proposed method will work in the absence of such data for generating the augmentations?

For MR reconstruction, the undersampling mask plays a significant role, while augmenting real images, what will be the effect on the mask. In particular, for 8x which masks are used in this work, are these similar to baseline or always used random sampling?

For results evaluation, SSIM is used, it is recommended to add results for either PSNR or NMSE for error quantification.

How are the samples for various data subsets selected?

Since it is claimed that this can be extended to other MRI datasets, is it possible to add results for Brain MRI as well fo having diverse data and hence a better representation on the generalisation of the MRAugment model. For model training, any form of transfer learning is used or all models are trained form scratch?

A comparison with state-of-the-art methods is missing and should be added.

Minor comments

There are typos and grammar related mistakes which should be corrected.

---

> ### Author Response · Authors · 2020-11-19
> **Response to Reviewer4**
>
> Thank you for taking the time reviewing our paper, your comments are highly appreciated! To address your concerns:
>
> **Re availability of fully sampled training data:** We completely agree that obtaining fully sampled data is time consuming and expensive and therefore we need to reduce the amount of necessary training data for reconstruction, which is one of the main focus of our paper. We would like to point out that learning-based reconstruction methods would always need fully sampled training data, since the ground truth reference is always fully sampled and without it there is no good way to validate the efficacy of any reconstruction technique. This is exactly why reducing the size of the training data is even more important as we can not accelerate acquisition when gathering training data. Therefore, MRAugment is always applicable where a supervised deep learning model is used. It is only at test time that we do not have fully sampled data.
>
> **Re undersampling masks:** We used the same random undersampling masks as in the fastMRI paper in order to provide fair comparison with the state-of-the-art. Other reviewers also pointed out that the mask is of interest and it has been added to the main part of the paper. In particular, we randomly mask full kspace lines in the phase encoding direction in order to make it easily realizable in a real scanner.
>
> **Re other metrics:** thank you for your comment on adding more evaluation metrics for comparison. We believe it is useful to provide a more comprehensive comparison of our results. Please find the PSNR plots in Appendix C.
>
> **Regarding sampling for various datasets:** again, this is a great point. We randomly sampled whole volumes of the full training dataset and added all slices corresponding to the sampled volumes. We added these details to the main part of our paper.
>
> **With respect to other datasets:** we agree with the reviewer that seeing results on the brain MRI dataset would be valuable. We are actively working on this, and the initial experiments are promising. However, we want to point out that a single baseline experiment (without data augmentation) on the complete brain dataset, due to its size, takes approximately 2800 GPU hours and requires at least 24GB GPU memory for training. Using our current academic setup, unfortunately this is not feasible. Furthermore, answering your question regarding transfer learning, the models were trained from scratch, no transfer learning applied. However, we have added a transfer learning experiment in Appendix F that suggests that data augmentation improves the robustness of the model to shifts in test distribution.
>
> **Regarding comparison to state-of-the-art:** we are unaware of any other comparable data augmentation techniques for accelerated MRI reconstruction, therefore we compared our results to state-of-the-art reconstructions on this dataset across all models. We are happy to compare in the final version of the paper if you have any specific papers/reconstruction techniques in mind.
>
> Thank you for pointing out the typos, we did a pass for typos on the full paper and will continue to check more carefully to fix any that we find.

---

### Official Review · AnonReviewer3 · 2020-11-02
**Recommendation to Reject**

**Rating:** 6
**Confidence:** 4

**Review:**

Example 2:


Review: This paper proposes data augmentation methods for medical imaging(especially for accelerated MRI) based on the MR physics. The augmentation includes both pixel preserving augmentations/general affine augmentations on both real and imaginary values in the image domain. Then, the augmented images are transformed to k-space domain and the k-space data are down-sampled for the input data generation for the accelerated MRI task. They claim that how to schedule p(the probability of applying combinations of augmentation) over the training is important and the schedules from p=0 and increasing over epochs shows best results, experimentally.

+ The importance of data augmentation is very important topic, especially in medical domain, because of their expensive cost for data gathering. It directly effects on the performance of deep learning algorithms.
+ Overall, the paper is well written and might be easy to read by the readers who are not familiar with accelerated MRI.
+ The results section is well structured. In particular, the results seem great and the proposed augmentation works well especially for the extreme cases such as using 1% of the training set.

Concerns:
- The key concern about the paper is the lack of experimentation to study the usefulness of the proposed method.
- Considering the limited results, a deeper analysis of the proposed method would have been nice. The idea of augmentation in the MR image domain for each real and imaginary domain is a generic one, and many of MR researchers are already tried to augment the MR image on image domain and then after down-sampling for the accelerated MR task. I cannot clearly see the novelty of the proposed algorithm except for scheduling.
-There are not enough ablation studies. For example, they are used variety of transformation for augmentation. The effects of each transformation should be different on the performance of the algorithm. Also, even the author divides the transforms into two categories: pixel preserving/general affine augmentation and the effects of two different augmentation can be helpful to the readers and researchers in the accelerated MRI deep learning community. However, it is missing.
- In the 3rd section, the author tries to link the necessary of the augmentation pipeline for preserving noise statistics with their algorithm. However, the analysis and comparison is not well-provided. I think the ‘naïve DA’ which they provided (just augmentation on real valued image) is not a fair comparison experiment since it is the half of the data (real vs complexed value). The results of the other augmentation algorithm and/or the more analysis based on their theory are missing.
- Also, the author mentioned about the physics of the data. However, all the augmentations they used(horizontal/vertical flips, rotations, translations, zoom-in/out, scaling, shearing) are the transformations which affects to the coil sensitivity map. Thus, the augmented train sets have the samples which have different coil sensitivity maps compared to the one which test sets have. I think they need to consider about these factors to claim about the physics based augmentation. (for example, 1. Calculate coil sensitivities of each coil, 2. Reconstruct the true magnetization X, 3. Apply augmentation on X and then apply the coil sensitivity map, 4. Get the k-space data and down-sampling.)
- The results seem more sharp then the one which not use the proposed domain adaptation and the SSIM score is better too. However, it is important the opinions of the clinician in department of radiology. In some cases, the clinicians might see some special features on the MR images even it is blurred. Thus, if the deep learning algorithm might lost that features, they prefer the blurred one even the other is more sharp. The evaluations by the clinicians make the author’s claim stronger.
- There should be some variation of sampling the train dataset (such as 1%). Multiple number of experiments and the presentation of the statistics of the results (mean/stddev)

Minor comments:
- The details about how they apply the scheduling of p(augmentation hyper-parameter) needs to be stated more.
-Data specification is missing which is important especially in the accelerated MRI. It should be stated in the supplementary materials.
- Not only for the sampling rate, but also details about the sampling pattern is important. Even if it is written in the paper of the model they used(End-to-End VarNet), it would be great to describe the details of the experiment.
- Is it work for the super-resolution task? I mean that even if the task was acceleration of MRI by super-resolution (the sampling pattern is low-pass filtering mask and the deep model is trained for that), does the augmentation successfully work?

---

> ### Author Response · Authors · 2020-11-19
> **Response to Reviewer3 - pt. 1**
>
> We are very grateful for your insightful and thorough review and we highly appreciate your effort reviewing our paper. Below we discuss the points you have raised in detail.
>
> **Re your concerns on novelty of data augmentation:** We agree that applying data augmentation to MR images in order to enlarge the training dataset has been investigated before in the context of image segmentation and classification. However, to the best of our knowledge, data augmentation for the MRI reconstruction problem has not been investigated at all. We would also like to highlight that data augmentation for reconstruction problems is fundamentally different from and significantly more challenging than data augmentation for classification problems. In particular, in reconstruction problems we have to carefully generate a pair of corresponding augmented target and augmented measurements, whereas in classification the class label is retained after augmentation. This is particularly challenging as for this to work we have to make sure the noise in the measurements has the same statistics with and without data augmentation (unlike classification problems) as when there is a noise mismatch DA can even hinder performance. Therefore, we believe that MRAugment provides significant novelty and utility to MR practitioners.
>
> **Re ablation studies:** The reviewer raised a very valid point regarding ablation studies. We agree that it would be interesting to see extensive experiments on the contribution of different transformations to the efficacy of MRAugment. However, training state-of-the-art models on the fastMRI knee dataset takes about 1200 GPU hours for a single experiment (!) with multi-coil data. We have access to an academic level of computational resources in the form of 8 GPUs to train these models, taking close to one week for a single experiment to finish. Therefore, it was not feasible for us to perform extensive ablation studies. However, to address the reviewer’s valid concern we verified the utility of the different augmentations on a subsampled dataset and observed that they are individually useful and their effect is complementary. We included these ablation studies in the Experiments section.
>
> **Re ‘naive data augmentation’:** We don’t simply discard half of the data and augment the real part. What we mean by naive is that we apply the augmentation to the target ground truth reconstruction, which is real-valued and obtained from the RSS of the complex multi-coil image. We call it naive, because most typical augmentations apply to simple real-valued images, and augmenting the target would be the most straightforward idea in the classical framework. We would like to emphasize that we do not mean to use naive data augmentation as an alternative or baseline to MRAugment. To the best of our knowledge there is no other data augmentation algorithm designed specifically for MRI reconstruction and therefore the baseline we use is training without data augmentation but with more/less data.
>
> **Re coil sensitivity map considerations:** The reviewer raised a very interesting point that we have also considered throughout our study. Namely, the transformations are applied to the underlying ground truth image modulated by the sensitivity map, and thus the sensitivity maps are also transformed. As the reviewer very correctly pointed out, we have two options:
> 1. Apply augmentation to the coil images including the coil as done in the paper. Even though the sensitivity maps are also transformed the model seems to be able to deal with these transformations as the experimental results are promising. The main benefit of this approach is that the noise distribution is not distorted as it would be using the other approach (see below) and we do not require estimated sensitivity maps for our algorithm to work.
> 2. We could also estimate coil sensitivities, recover the underlying ground truth object and apply data augmentation. Then, modulate the augmented object with the different estimated sensitivities. This method has a much larger distorting effect on the noise distribution and introduces correlation between augmented images corresponding to different coils.  We added a detailed discussion in Appendix E with the derivation. This mismatch in the noise distribution can degrade performance if not accounted for. As mentioned in the paper we are working to enhance MRAugment to be able to handle such mismatches in the future in which case such a strategy may become feasible. However, without any adjustment we believe that it will have a detrimental effect and have therefore avoided this in our current setup.

---

> > ### Author Response · Authors · 2020-11-19
> > **Response to Reviewer3 - pt. 2**
> >
> > **On the need for clinician opinion:** We agree with the reviewer, an expert opinion on the diagnostic value of reconstructions is important and we are actively working on obtaining feedback from a board of radiologists. Our preliminary consultation with an MSK radiologist at the University of Utah indicates that our reconstructions are indeed acceptable for diagnostic purposes.
> >
> > **On minor comments:**
> > - Please find details of the exact scheduling functions we used in Appendix B. For more implementation specific details please refer to the source code attached to the main submission. Please let us know if there are further important details with respect to scheduling that you are interested in.
> > - Could you please elaborate more on what you mean by data specification? We used the fastMRI dataset, of which details can be found in the respective paper. A short overview of this dataset is provided in Appendix A.
> > - Thank you for pointing out the importance of the undersampling mask. We used the random mask specified in the fastMRI paper and we added a description of the mask in the main part of our paper. In particular, we randomly mask full kspace lines in the phase encoding direction in order to make it easily realizable in a real scanner.
> > - It is an interesting question whether MRAugment would be effective for super-resolution and something definitely worth trying. We are excited to broaden the possible range of applications where MRAugment is useful.

---

> > ### Comment · AnonReviewer3 · 2020-11-24
> > **I have few questions as followings.**
> >
> > Dear Authors,
> > Thank you for the kind replies and explanations. I have few questions as followings.
> >
> > For the MR reconstruction task in deep learning, 'data augmentation in complex image domain' is already tried by many researchers [1,2] not only in the context of image segmentation and classification but also MR acceleration task.
> > [1] SANTIS: Sampling-Augmented Neural neTwork with Incoherent Structure for MR image reconstruction, Liu et al. MRM (2019)
> > [2] Deep Residual Learning for Accelerated MRI Using Magnitude and Phase Networks, Lee et al.  IEEE Trans. BME (2018)
> > They transformed the k-space to complex image domain and augmentation in real and imaginary images and then downsampled to generate pair of target (full-sampled k-space) and input (down-sampled k-space) for MR acceleration tasks. Therefore, I think the proposed way to augment is not novel except for the scheduling, and your contribution is formulation of the problem with analysis about noise distribution.
> >
> > In this manner, I think the manuscript should elaborate more about the point as followings: 'preserving noise statistics’, ‘augmentation performance compare to the other algorithm e.g. SANTIS’,  ‘How much applying the sampling mask for every epoch randomly affect to the performance’,
> >
> > Did the hyper-parameter for the scheduling (p) find on validation set?
> >
> > When you divide the train and validation sets, is there any performance difference when you use different random seed for the sampling?

---

> > > ### Author Response · Authors · 2020-11-25
> > > **Thank you for the follow-up, see below our response**
> > >
> > > Thank you for your comment, we really appreciate your time reviewing our paper and helping us with insightful comments. We are pleased to see that we have managed to address most of your initial concerns. Please find below our response to your comments.
> > >
> > > **Re [1]:** The paper you have cited (SANTIS) uses randomly generated undersampling masks as a form of regularization against undersampling artifacts. As we mentioned it in Appendix B, *we already use this technique* in training both the baseline models and the models with our data augmentation pipeline. We would like to point out that this is a fairly standard technique in deep learning based MRI reconstruction, and all competitive models on the fastMRI dataset use this technique by default. Since the baselines are already trained using randomly generated undersampling masks during training, and these are the best performing models on the dataset, we believe comparison with SANTIS would not be too informative. Our contribution on data augmentation is closer to the notion of data augmentation in standard deep learning literature (7.4 in [3]). However, we would like to make it more clear in our paper that all models were trained using randomized undersampling masks, therefore we clarified it in the main body of the paper (not only in Appendix). Thank you for pointing this out!
> > >
> > > **Re [2]:** This paper focuses on training separate networks for reconstructing the magnitude and phase images separately. We are aware that treating real/imaginary or magnitude/phase parts separately in deep learning based MR reconstruction is standard and we do not claim it as a novelty of our paper. However, we believe that an effective data augmentation pipeline for accelerated MRI reconstruction has not been thoroughly explored in the literature. We realize that [2] applied random flips to enlarge the training set by a factor of 4, but data augmentation is definitely not the focus of [2] and they did not investigate the effect of data augmentation at all. Therefore, we believe that our paper still provides significant novelty and contribution compared to prior work. However, to provide a more complete picture, we are happy to mention [2] when discussing related work. Thank you for bringing this paper to our attention!
> > >
> > > **Re tuning p:** You are correct that the results presented in Table 2 on the effect of augmentation probability are evaluated on the validation dataset. We made it more clear in the main body of our paper.
> > >
> > > **Re dividing train and validation datasets:** The fastMRI dataset has a fixed validation dataset separate from the training dataset in order to make sure the results are completely comparable across different papers. As we mention it in the *Experimental setup* part of our paper, we used this fixed validation dataset in our experiments. We did not generate different train/val splits from the training dataset in order to maintain compatibility with prior results on the dataset.
> > >
> > > [1] SANTIS: Sampling-Augmented Neural neTwork with Incoherent Structure for MR image reconstruction, Liu et al. MRM (2019)
> > >
> > > [2] Deep Residual Learning for Accelerated MRI Using Magnitude and Phase Networks, Lee et al.  IEEE Trans. BME (2018)
> > >
> > > [3] Goodfellow, I., Bengio, Y., Courville, A. and Bengio, Y., 2016. *Deep learning*. Cambridge: MIT press.

---

### Official Review · AnonReviewer5 · 2020-11-08
**Data augmentation for deep learning based accelerated MRI reconstruction**

**Rating:** 6
**Confidence:** 4

**Review:**

This paper gives a simple yet straightforward method to do data augmentation in MRI imaging. The paper is well written and the results quite promising on a rigorous test case.

However, the methods proposed are in a sense rather obvious and the results do not give an improvement on the main practical test case. Throwing away data-post hoc sadly isn’t as strong of a motivation as showing an improvement on the full dataset.

Still, the methods are novel, the presentation good and all steps are done rigorously so I can certainly see this being useful for a huge range of related applications. For this reason I’ll recommend acceptance.

Some further comments include:

* As noted by the authors, post-processing methods are an alternative to physics based approaches. An upside of them is that it’s trivial to do very aggressive data-augmentation (see e.g. Deep Bayesian Inversion which applied the whole menagerie). It could be interesting to have a standard U-Net baseline.
* The “MRI is a Fourier transform” argument is well known to be a rather coarse approximation. It would be interesting to hear some debate on how a slight error in the forward model would impact this application.
* Related to the above, MRI is a “trivial” inverse problem since the Fourier transform is unitary, basically a “rotation” in some high dimensional space. Comments on how to apply these methods to more advanced inverse problems like CT would be most welcome.
* H and V flip looks wrongly labeled in Fig2.
* How are the images e.g. rotated without padding artefacts?
* In figure 12 the baseline should also show a training curve
* How are the windows selected in e.g. figure 10? It feels like it’s slightly different between samples and also a bit too wide.
* All figures that are not “images” should be in vector format. Notably figure 4.

---

> ### Author Response · Authors · 2020-11-19
> **Response to Reviewer5**
>
> We appreciate your effort in reviewing our paper and thank you for the helpful comments. We thoroughly address your concerns below.
>
> **Re MRAugment not useful on the full dataset**: We would like to point out that the main goal of our paper is to demonstrate how data augmentation can help in the low data regime for MRI reconstruction (which overwhelmingly arises in practice) and *not* to exceed the fastMRI baseline on the full dataset. In fact the reason we used a large dataset such as the fastMRI dataset is to analyse the impact of the amount of training data with and without data augmentation. We want to emphasize that in most MRI applications such a large dataset is not available for training. Therefore, there is a stronger motivation to improve low-data reconstruction performance than improving a baseline for which the amount of training data is rarely met in practice. A good example of this is low-intensity magnetic field MRIs which show a lot of promise in making MRI safer and applicable to many new settings, including patients with pacemakers or metallic inserts. However, there are only three in the entire US dedicated to a wide variety of research areas so that it is not feasible to gather such large datasets. Even in the high-intensity regime, such as the fastMRI data, it is not clear that the conclusions generalize to other MR machines without calibration and it may not be feasible to gather such a large training data for each new machine immediately. That said, we believe that with increased network capacity (add more layers or make the network a bit wider), by using data augmentation we could even improve upon the state-of-the-art as DA is most helpful in a regime where the network capacity is so large that without DA the network overfits. However, this is currently out of the scope of this paper as each experiment on the entire dataset takes almost a week to train on our 8-GPU server and therefore overparameterization by a factor of even 4 requires training for almost a month, which is currently not feasible for us. However, we are in the process of purchasing more GPUs and we are optimistic that we can eventually improve over the entire dataset. Additionally, we do observe a benefit of MRAugment on the full training dataset in the form of improved performance on new datasets not seen by the model. We added further details on this in Appendix F.
>
> **Re Fourier transform is a coarse approximation**: thank you for bringing this to our attention, could you please provide some references? To the best of our knowledge, Fourier transform is justified by the physics of the acquisition process as there is a one-to-one mapping between spatial locations and the resonance frequency of the excited spins. We are aware that there are some simplifying assumptions, such as the neglected T2 decay during readout, however this is considered standard in the literature. Moreover, we are confident that deep learning based reconstruction is fairly robust to these slight disturbances in the forward model (please see next point).
>
> **Re perturbations in the forward model and more complex inverse problems:** We believe that the proposed method is not too sensitive to variations in the forward model and could be applied to more ill-posed operators such as CT. Successful deep learning models in medical image reconstruction typically exploit the UNet architecture in one way or another. This architecture provides an excellent natural image prior and a form of implicit regularization that automatically corrects for small variations in the forward model and provides implicit regularization to more ill-posed problems such as CT. Data augmentation exploits spatial invariances that are present in any naturally occurring medical images and thus we are positive that MRAugment can be applied in a broader context.
>
> We did not understand your point on post-processing approaches as an alternative to physics based approaches. Could you please elaborate what you mean?
>
> **Comments regarding figures:**
> -Thank you for pointing out that the captions for horizontal and vertical flipping are switched, we corrected it.
>
> -When applying the augmentations such as rotation we use reflection padding. The target images are center-cropped and therefore the padded region is rarely visible.
>
> -Thank you for the advice, we included  the training curve for Fig. 12.
>
> -The windows are 320x320 center crops across the paper consistently with the target images in the fastMRI dataset. The apparent width difference is due to the varying depth of the slice: anatomies on center slices are wider than slices close to the bottom or top of the volume.

---

### Decision · Program_Chairs · 2021-01-07
**Final Decision**

**Decision:**

Reject

**Comment:**

Reviewers could not reach consensus here and legitimate concerns are raised on novelty and on empirical results, although this can be attributed to the important computation times required to run experiments on 3D MRI volumes. The authors have provided a comprehensive response to the reviews, the general feedback is that the work has merit but it fails to convince on its real contribution to the state-of-the-art. At this stage, I fear this work cannot be recommended for acceptance.